# Recognition of Dynamic Emotional Expressions in Children and Adults and Its Associations with Empathy

**DOI:** 10.3390/s24144674

**Published:** 2024-07-18

**Authors:** Yu-Chen Chiang, Sarina Hui-Lin Chien, Jia-Ling Lyu, Chien-Kai Chang

**Affiliations:** 1School of Medicine, China Medical University, Taichung 404, Taiwan; u106001433@cmu.edu.tw; 2Graduate Institute of Biomedical Sciences, China Medical University, Taichung 404, Taiwan; u105306601@cmu.edu.tw (J.-L.L.);; 3Center for Neuroscience and Brain Disease, China Medical University, Taichung 404, Taiwan

**Keywords:** face perception, dynamic emotion recognition, empathy quotient (EQ), motor empathy, school-age children

## Abstract

This present study investigates emotion recognition in children and adults and its association with EQ and motor empathy. Overall, 58 children (33 5–6-year-olds, 25 7–9-year-olds) and 61 adults (24 young adults, 37 parents) participated in this study. Each participant received an EQ questionnaire and completed the dynamic emotion expression recognition task, where participants were asked to identify four basic emotions (happy, sad, fearful, and angry) from neutral to fully expressed states, and the motor empathy task, where participants’ facial muscle activity was recorded. The results showed that “happy” was the easiest expression for all ages; 5- to 6-year-old children performed equally well as adults. The accuracies for “fearful,” “angry,” and “sad” expressions were significantly lower in children than in adults. For motor empathy, 7- to 9-year-old children exhibited the highest level of facial muscle activity, while the young adults showed the lowest engagement. Importantly, individual EQ scores positively correlated with the motor empathy index in adults but not in children. In sum, our study echoes the previous literature, showing that the identification of negative emotions is still difficult for children aged 5–9 but that this improves in late childhood. Our results also suggest that stronger facial mimicry responses are positively related to a higher level of empathy in adults.

## 1. Introduction

Various aspects of facial recognition (e.g., identity, sex, race, and age) play an important role in our social functioning and interpersonal relationships. Among these, correctly reading facial expressions is essential for successful interaction with others and crucial for developing the Theory of Mind (ToM) [1,2]. Spontaneous preferences for faces [3,4,5] and sensitivity to basic facial emotions emerge early in infancy [6,7]; however, the recognition and understanding of emotional expressions continue to develop and are refined from preschool to middle childhood and adolescence [8,9,10,11,12,13].

Using static photographs or cartoons depicting the full intensity of the six basic emotions (happiness, sadness, fear, anger, disgust, and surprise), Kolb et al. [12] reported that emotion recognition improved between 6 and 10 years of age and improved again between 14 to 15 years of age and in adulthood. Mondloch et al. [13] showed that the accuracy in matching an emotional photograph to one of four expressions (neutral, surprise, happiness, or disgust) increased from age 6 to 8 and that the developmental pattern was not uniform across the six basic emotions. While the expressions of happiness and sadness seem to be correctly categorized earlier than those of fear and disgust, the findings for anger expression are less consistent [14,15,16,17,18].

Studies using varying intensities of facial expressions further confirmed the non-uniform development across emotional expressions. Herba et al. [10] examined emotion processing in a large sample of 4- to 15-year-old children and found that the accuracy in recognizing sadness improved at a lower rate than for happiness, fear, and disgust. Gao and Maurer [19] tested 5- to 10-year-olds with various intensities of emotional expressions and found that 5-year-olds were as sensitive as adults in recognizing happiness, even at low intensities. For sadness, 5-year-olds could judge that the face was expressive (i.e., not neutral); however, even at 10, children were still likely to misjudge sadness as fearful. The development of fear-related expressions appeared more slowly; children’s thresholds reached adult-like levels at age 10.

Natural human interactions are highly dynamic; compared to static images, dynamic faces are more ecologically valid depictions of how expressions are encountered in real-life situations. Using dynamic emotion expression stimuli (short video clips of faces moving from neutral expression to full intensity), Montirosso et al. [18] investigated the development of the six basic emotions in 4- to 18-year-old children and adolescents and reported that sadness and anger were the least accurately recognized expressions. In another large cross-sectional study conducted using static and dynamic emotional expressions across the life span (5–96 years old), Richoz et al. [11] showed that fear recognition had the sharpest improvement from childhood to early adulthood, followed by the recognition of disgust and surprise. These findings align with previous studies showing that the recognition of fear and disgust significantly improves with age [10]. A more gradual increase in the recognition of sadness and anger was observed in the static condition.

In addition to developmental maturation, factors such as gender, IQ, the efficiency of encoding faces [20], and socioeconomic status (SES) have been reported to modulate the efficacy of emotion processing [8]. However, research investigating the link between empathy—“an affective response more appropriate to someone else’s situation than to one’s own” [21]—and emotion recognition remains scarce. Choi and Watanuki [22] found that adults with higher empathy scores showed better performance and deeper involvement in a facial expression recognition task. Using eye tracking with odd-one-out and pain evaluation tasks, Yan et al. [23,24] revealed a significant link between empathy and the perception of facial pain in preschoolers and adults. The cohort of 5–6-year-olds detected painful expressions faster than other expressions (angry, sad, and happy); children and adults with high degrees of empathy were better at detecting painful expressions and tended to give higher subjective pain intensity ratings.

Although Preston and de Waal [25] regard “empathy” as a unitary process, a super-ordinate category including all sub-classes of phenomena that share the same core mechanism (emotional contagion, sympathy, cognitive empathy, etc.), others have argued that the term “empathy” subsumes a set of dissociable neurocognitive processes with three main divisions: cognitive, motor, and emotional empathy [26]. For example, motor empathy is the tendency to automatically mimic and synchronize facial expressions, vocalizations, postures, and movements with another person [27]. It has long been considered a “primitive form of sympathy” [28,29], the ability to understand and share the emotions of others by simulating the experiences in one’s body. A neurocognitive account based on mirror neurons, which show activity during the execution and the observation of action [30], has been developed to support the idea of motor empathy [31,32]. Several theories have tried to link tendencies to mimic others’ facial movements to empathy and facial emotion recognition, although evidence for such links is scarce. A recent meta-analysis study [33] summarized that stronger facial mimicry responses are positively related to a higher level of empathy but not to facial emotion recognition, and argued that this association is conditional on factors not fully understood.

The relationship between empathy, motor empathy, and emotion recognition remains unclear in adults and relatively unexplored in children. Therefore, the present study aims to understand the interplay (or the lack of it) among empathy, motor empathy, and emotion recognition in children and adults. Specifically, we investigated the performance of basic emotion recognition, the empathy quotient (EQ), and motor empathy in 5- to 9-year-old children, young adults, and middle-aged adults who were also parents. Each participant received an EQ questionnaire and two computerized tasks. The first task was dynamic facial emotion recognition, where participants were required to identify four basic emotions (happy, sad, fearful, and angry) in video clips of faces moving from a neutral state to a fully expressed emotional state. We employed dynamic faces because they are ecologically more valid and closer to the expressions encountered in real life. The second task assessed an individual’s spontaneous motor empathy. Participants watched three short video clips (selected from YouTube) with easily identifiable emotional content, which were simultaneously recorded and scored by an emotion expression coding software—iMotions(10)™ Affectiva.

## 2. Materials and Methods

### 2.1. Participants

A total of 58 children and 61 adults participated in this study. The child participants were divided into the 5–6-year-old group (*n* = 33, *M* = 6.34 yrs *SD* = 0.6; 15 boys, *M* = 6.52, *SD* = 0.58; 18 girls, *M* = 6.19, *SD* = 0.59) and 7–9 year-old group (*n* = 25, *M* = 8.08, *SD* = 0.72; 13 boys, *M* = 8.04, *SD* = 0.78; 12 girls, *M* = 8.12, *SD* = 0.68). The adult participants were divided into the young adult group (*n* = 24, 12 females, 12 males) with a mean age of 23.02 ± 2.79 yrs, and the parent’s group (*n* = 37, 35 females, 2 males) with a mean age of 37.29 ± 4.96 yrs. Children were recruited from the Taichung Metropolitan area via the China Medical University InfantLab Child Participant Database. The participants of the young adult group were recruited from China Medical University via The China Medical University Joint Psychological Experiment Participant Recruitment Website (http://www2.cmu.edu.tw/~psychology/ (accessed on 14 November 2020)) and the Student Facebook club. The participants in the parents’ group were those parents of child participants who were willing to join the study. Each participant, either adult or child, had normal or corrected-to-normal vision (20/20). This present study adhered to the ethics of scientific publication as detailed in the Ethical Principles of Psychologists and the Code of Conduct [34], and the study protocol was approved by the Institution Review Board of the Research Ethics Committee at China Medical University Hospital, Taichung, Taiwan (Certificate number: CMUH103-REC3-055). The adult and child participants’ parents were informed of the study’s general purpose and experimental procedures. We obtained informed written consent (for adult participants) or the informed parental consent of the child participants before the study. Each adult and child participant received (1) the Chinese version of the empathy quotient (EQ) questionnaire, (2) the dynamic facial emotion recognition task, and (3) the motor empathy task in a fixed order. The total duration of the study was about 30 min. Adult participants received cash compensation at the end of the study. Child participants received cash compensation, a certificate of participation, and a small gift for their visit.

### 2.2. The Chinese Version of the Empathy Quotients (EQ) Questionnaire

#### 2.2.1. Versions

Adult participants self-evaluated themselves in Chinese using the adult empathy quotient (EQ) questionnaire by Sher et al. [35], which was adopted from Baron-Cohen and Wheelwright [36]. Among the 68 original questions, the Chinese version included 40 questions that were translated idiomatically by language professionals. Children were assisted by their parents in completing the child empathy quotient (EQ) questionnaire in Chinese using the version by Huang [37], which was adopted from Auyeung et al. [38]. The Chinese version deleted inadequate, culturally bound questions in the Taiwanese context, resulting in 32 questions. Among these, 16 questions evaluated children’s level of empathy, and 16 questions assessed their level of systemizing.

#### 2.2.2. Procedures

We used SurveyCake (developed by 25 Sprout, Taipei) to compile the Adult EQ and Child EQ questionnaires into online versions. Participants could answer the questionnaires with their laptops or smartphones. There are four options for each question in the Adult EQ and Child EQ: agree entirely, slightly agree, slightly disagree, and completely disagree. Completely agree scores 2 points, slightly agree scores 1 point, and slightly disagree and completely disagree score 0. For the reverse questions, completely disagree scores 2 points, slightly disagree scores 1 point, and slightly and completely agree score 0. According to Baron-Cohen and Wheelwright [36], the average scores of British men and women were 41.8 ± 11.2 and 47.2 ± 10.2, respectively. According to Huang [37], the average scores of the empathy part of the Chinese version of Child EQ for Taiwanese boys and girls were 20.21 ± 7.05 and 23.08 ± 6.77, respectively.

### 2.3. Dynamic Facial Emotion Recognition Task

#### 2.3.1. Apparatus and Stimuli

A 15.6” laptop computer (Acer P59-M5726) and E-Prime 2.0 (Psychology Software Tools, Sharpsburg, PA, USA) were used to run the task and to record the participants’ responses. We used neutral, angry, fearful, happy, and sad expressions for the stimuli of one female face and one male face (from the Taiwan Facial Expression Image Database TFEID [39]). Using FantaMorph 5 Deuluxe (Abrosoft Co., Pasadena, CA, USA), we created 12 color dynamic facial emotion expression GIF videos by morphing the neutral face (considered as 0% intensity) with each of the four basic emotions in their full-intensity states (angry, fearful, happy, sad—100% intensity) for each gender (please see Ho et al. [40]; Chien et al. [41]; and Chen et al. [42] for the detailed procedures involved in creating the dynamic morphing emotional face stimuli). Each dynamic facial emotion expression GIF video started from a neutral state and naturally evolved into a full expression at the frame rate of 30 frames per second. Each dynamic facial emotion video clip was approximately 2000 milliseconds, and the dimensions of the face images on the screen were about 17.3 cm high × 15.2 cm wide. This task contained 48 video clip trials (4 emotions × 2 genders × 6 repetitions), which were presented randomly (see Figure 1: left panel).

#### 2.3.2. Procedures

Each participant sat on an adjustable chair, allowing their eyes to fixate on the center of the screen with a viewing distance of about 30 cm. Each trial began with 1 s of blank screen, followed by a video clip showing dynamic facial emotions evolving from a neutral state into the most intense expression. The video clip was programmed to play for 2 s automatically. Adult and child participants were told to make a 4AFC keypress response (four response keys labeled “Angry”, “Fearful”, “Happy”, and “Sad”) as fast and accurately as possible when they recognized the emotion being expressed. When the participant made a keypress response while the video was still playing, the video clip stopped immediately, their response accuracy and reaction time were recorded, and the next trial began. When the participant made a keypress response after the video clip was completed, the screen stayed at the last frame of the video (full intensity of the expression), the participant’s response accuracy and reaction time were recorded, and the next trial began. To familiarize the participants with the four corresponding response keys for the four basic emotions, each participant received 16 practice trials before the formal experiment. The practice trials employed static face images of “Angry”, “Fearful”, “Happy”, and “Sad” expressions of full intensity (4 facial expressions × 2 genders × 2 repetitions), and the face images in the practice session were not included in the formal experimental trials.

### 2.4. Motor Empathy Task

#### 2.4.1. Apparatus and Stimuli

A 15.6” laptop computer (Acer P59-M5726) was used to run the task. We utilized i-Motion Affectiva (i-Motions, Ltd., Copenhagen, Denmark) to record and analyze participants’ spontaneous facial expressions in real time. The video clips were selected from YouTube. The most important criterion was that the main character in each clip expressed an easily recognized expression that lasted for about 10 s. We originally picked eight clips for pilot testing to make sure that participants could easily understand the emotions in the video. We selected three final video clips with sad (clip#1, 50 s, https://www.youtube.com/watch?v=2qaSSdtyvGc&t=463s) (accessed on 14 November 2020)”., nausea (clip#2, 24 s, https://www.youtube.com/watch?v=oQYPgD3cIP0&t=586s) (accessed on 14 November 2020), and happy (clip#3, 31 s, https://www.youtube.com/watch?v=4shUUjcjUx8&t=157s)(accessed on 14 November 2020) scenarios as the primary stimuli.

#### 2.4.2. Procedure

Participants sat on an adjustable chair, allowing their eyes to fixate on the center of the screen at a viewing distance of 30 cm. Participants were asked to passively watch the three short videos (total length = 105 s); their spontaneous facial movements and expressions were recorded live via the built-in laptop camera. While the videos were playing, each participant’s facial expression recordings were analyzed using iMotion Affectiva (facial expression recognition engine) software (see Figure 1: right panel). The software detects facial landmarks and classifies facial expressions in return, with numeric scores for facial expressions, Action Unit (AU) codes, level of engagement, and other metrics. The scores range from 0 (no expression) to 100 (expression fully present).

## 3. Results

### 3.1. Chinese Empathy Quotient

Because the adults and the children used different versions of the Chinese empathy quotient, we analyzed the adults and the children’s EQ scores separately (see Table 1). For adults, the group mean EQ scores for young adults and the parents were 41.58 (*SD* = 14.15) and 47.89 (*SD* = 9.93), and the difference was significant (*t*(49) = 2.056, *p* = 0.045). To determine whether there was a sex difference, we conducted two separate independent *t*-tests for the young adults and the parents. For young adults, the EQ scores for males (*M* = 42.17, *SD* = 17.61) and females (*M* = 41.00, *SD* = 10.40) were not significantly different (*p* = 0.845). For parents, the EQ scores for males (*M* = 36.00, *SD* = 7.95) and females (*M* = 48.57, *SD* = 9.63) were marginally significantly different (*t*(35) = 1.794, *p* = 0.081).

For children, the group mean EQ scores for the 5–6-year-olds and 7–9-year-olds were 18.42 (*SD* = 7.36) and 18.68 (*SD* = 7.78), which were not significantly different (*p* = 0.899). We also conducted two independent *t*-tests to reveal whether there was a sex difference in children. For the 5- to 6-year-old group, the EQ scores for boys (*M* = 18.27, *SD* = 8.56) and girls (*M* = 18.56, *SD* = 6.45) were not significantly different (*p* = 0.913). For the 7–9-year-old group, the EQ scores for boys (*M* = 17.54, *SD* = 6.77) and girls (*M* = 19.92, *SD* = 8.88) were also not significantly different (*p* = 0.457).

### 3.2. Dynamic Facial Emotion Recognition

To test whether there was an effect of age group and the types of emotional expressions, we conducted two separate 2-way mixed ANOVAs on accuracy and response time, with *Group* (5–6-year-old, 7–9-year-old, young adults, parents) acting as the between-subject factor, and *Emotion Type* (angry, fearful, happy, sad) as the within-subject factor. For accuracy, the *group* main effect was significant (*F*(3, 115) = 13.160, *p* < 0.001, *η_p_*^2^ = 0.256); from high to low, the mean accuracies for parents, young adults, 5–6 year-olds, and 7–9 year-olds were 0.938 (*SE* = 0.016), 0.901(*SE* = 0.020), 0.817 (*SE* = 0.020), and 0.808 (*SE* = 0.017), respectively. Post hoc Scheffe tests (with an adjusted α level = 0.05/6 = 0.008) revealed that the mean accuracy of 5–6-year-olds was significantly lower than that of the parents (*p* < 0.001) and that of the young adults (*p* = 0.001). The mean accuracy of 7–9-year-olds was also significantly lower than that of the parents (*p* < 0.001) and that of the young adults (*p* = 0.004). However, there were no significant differences between the two groups of children and the two groups of adults. The main effect of *Emotion Type* was significant (*F*(3,115) = 29.526, *p* < 0.001, *η_p_*^2^ = 0.204). From highest to lowest, the mean accuracies for happy, sad, angry, and fearful expressions were 0.993 (*SE* = 0.022), 0.858 (*SE* = 0.192), 0.823 (*SE* = 0.202), and 0.805 (*SE* = 0.262), respectively. Post hoc Scheffe tests (with an adjusted α level = 0.05/6 = 0.008) revealed that the differences between happy and angry (*p* < 0.001), happy and fearful (*p* < 0.001), and happy and sad (*p* < 0.001) were all significant. The difference between fearful and sad (*p* = 0.043) was marginally significant. The rest of the pairwise comparisons were not statistically significant.

More importantly, the *group * Emotion Type* interaction was significant, where (*F*(9,345) = 2.675, *p* = 0.005, *η_p_*^2^ = 0.065). We further analyzed the group simple main effect for each Emotion Type with one-way ANOVA. For ‘happy’, the group simple main effect was not significant (*p* = 0.406), indicating that participants of all ages performed equally well. For ‘angry’, the group simple main was significant (*F*(3, 114) = 8.389, *p* < 0.001). With an adjusted α = 0.05/6 = 0.008, the parents performed better than 5–6 years old (*p* = 0.004) and 7–9 years old (*p* = 0.003). For ‘fearful’, the group simple main effect was significant (*F*(3, 114) = 5.656, *p* = 0.001); with an adjusted α = 0.05/6 = 0.008, the parents performed better than the 5–6 years old (*p* = 0.006) and marginally better than 7–9 years old (*p* = 0.06). For ‘sad’, the group simple main was also significant (*F*(3, 114) = 3.368, *p* = 0.021); the parents performed marginally better than the 5–6-year-old (*p* = 0.014). Figure 2 illustrates the group mean recognition accuracies for the four emotion expressions in children and adults.

For the response time (RT), only the correct trials were included in the analysis. The results of the two-way mixed ANOVA on response time (in milliseconds) showed that the group main effect was significant (*F*(3, 111) = 15.80, *p* < 0.001, *η_p_*^2^ = 0.290). From fast to slow, the mean RTs for young adults, parents, 7–9-year-olds, and 5–6-year-olds were 2072 ms (SE = 135.77), 2225 ms (*SE* = 108.67), 2391 ms (*SE* = 135.77), and 3126 ms (*SE* = 115.11), respectively. Post hoc Scheffe tests (α = 0.05/6 = 0.008) revealed that the mean response time of 5–6 years olds was significantly slower than that of 7–9 years olds (*p* < 0.001), that of young adults (*p* < 0.001), and that of parents (*p* < 0.001). However, the differences in RT among 7–9-year-olds, young adults, and parents were not significant. The main effect of emotion type was significant (*F*(3, 111) =87.184, *p* < 0.001, *η_p_*^2^ = 0.440). From fast to slow, the mean RTs for happy, angry, sad, and fearful were 1747ms (*SE* = 42.55), 2457 ms (*SE* = 77.66), 2735 ms (*SE* = 91.43), and 2832 ms (*SE* = 91.98), respectively. Post hoc Scheffe tests (α = 0.05/6 = 0.008) revealed that the response time of happy and angry was significantly faster than all other emotion expressions (*p* < 0.001 for all emotions). The group * Emotion Type interaction was not significant (*p* = 0.092). Figure 3 illustrates the group mean RTs for the four emotion expressions in children and adults.

### 3.3. Motor Empathy

To measure motor empathy in all age groups, we used the iMotions Affectiva software to record participants’ spontaneous facial reactions when viewing the three video clips in real-time. The software detects facial landmarks and classifies the facial expressions in return with numeric output scores for the level of engagement and several facial emotion categories. Here, we employed the maximum engagement score to represent the motor empathy level [40]. Because the range of individuals’ original engagement scores across different video clips expanded over 4 to 5 log units, we applied logarithm transformation (log of 10) to the original scores. Hence, we used log values (Engagement Max) as the dependent variables and then conducted a 2-way mixed ANOVA with *groups* and *Video Clip* on the log (Engagement Max) scores. The *group* main effect was significant, with *F*(3, 114) = 4.326, *p* = 0.006, and *η_p_*^2^ = 0.102. From high to low, the group mean log (Engagement Max) for 7–9-year-olds, 5–6-year-olds, parents, and young adults were 1.776 (*SE* = 0.130), 1.506 (*SE* = 0.113), 1.450 (*SE* = 0.113), and 1.081 (*SE* = 0.133), respectively. The *Video Clip* main effect was not significant (first clip: *M* = 1.489, *SE* = 0.079; second clip: *M* = 1.356, *SE* = 0.088, third clip: *M* = 1.514, *SE* = 0.070. *p* = 0.122).

Importantly, the *group* * *Video Clip* interaction was significant (*F*(6, 228) = 2.724, *p* = 0.014, *η_p_*^2^ = 0.067). To further explore the effect of age on each clip, we conducted three one-way ANOVAs for each video clip. For the first clip, the group simple main effect was not significant (*p* = 0.141). For the second clip, the group simple main effect was significant (*F*(3, 114) = 3.542, *p* = 0.017). Post hoc Scheffe tests (α = 0.05/6 = 0.008) revealed that the log maximum engagement score of 7–9-year-olds and young adults was significantly different (*p* = 0.004). For the third clip, the group simple main effect was also significant (*F*(3, 114) = 6.458, *p* < 0.001). Post hoc Scheffe tests (α = 0.05/6 = 0.008) showed that the young adults had significantly lower log maximum engagement scores than those of the 5–6 year (*p* = 0.001), the 7–9 year (*p* = 0.006), and the parent group (*p* = 0.005). Figure 4 illustrates the group mean log (Engagement Max) for the three video clips in children and adults.

### 3.4. Correlations among EQ, Task Performance, and Motor Empathy

We conducted Pearson’s correlations to explore the associations among empathy (EQ score), dynamic facial emotion recognition performance (accuracy and response time), and motor empathy (log maximum engagement). Because the adults and the children used different EQ measures with different maximum scores, we thus conducted separate correlational analyses for adults (including young adults and parents) and children (including 5–6- and 7–9-year-olds). Table 2 summarizes the correlation strengths and *p*-values for adults and children.

For adults, the EQ score did not correlate with the performance in dynamic facial emotion recognition. However, we observed positive correlations between EQ and the log (Engagement Max) of the second (*r* = 0.312, *p* = 0.015) and the third (*r* = 0.405, *p* = 0.001) clips, indicating that the adults with higher empathy scores tended to show stronger emotional engagement when viewing the second and the third clips. We also observed some significant correlations among the performance indices of the dynamic facial emotion recognition task. For example, the accuracies for “fearful” and “sad” (*r* = 0.429, *p* = 0.001) were positively correlated, meaning that those who were more accurate at identifying “fearful” expressions were also more accurate at identifying “sad” expressions. Similarly, the accuracies for “anger” and “happy” (*r* = 0.271, *p* = 0.037) were positively correlated, meaning that those who performed better at recognizing “angry” expressions were also better at the “happy” condition. Interestingly, the accuracy of “sad” correlated negatively with the reaction times of “happiness” (*r* = −0.265, *p* = 0.041) and “sad” (*r* = −0.358, *p* = 0.005) expressions, suggesting that those who were more accurate at identifying “sad” expressions also tended to recognize the “happy” and “sad” expressions faster. Moreover, the adults who performed better with the “sad” condition during the dynamic facial emotion recognition also showed greater engagement when watching the third video clip (with the central theme of happiness). Lastly, the reaction times for the different dynamic facial emotion expressions positively correlated with one another, indicating a consistency in the individual’s speed of response. Similarly, the log (Engagements Max) of the three video clips also positively correlated with one another.

For children, the EQ score positively correlated with the accuracy of the “happy” condition (*r* = 0.307, *p* = 0.019), indicating that those children who had a higher EQ score were more accurate at recognizing “happy” expressions. However, unlike adults, children’s EQ score did not correlate with motor empathy indices. Nevertheless, we observed some associations between the reaction times of dynamic emotion recognition and motor empathy. The log (Engagement Max) of the first clip negatively correlated with the reaction time for “angry” (*r*= −0.268, *p* = 0.042), “happy” (*r* = −0.276, *p* = 0.036) and “sad” (*r*= −0.268, *p* = 0.042) conditions, meaning that those children who were faster at identifying “anger,” “happy” and “sad” expressions tended to show stronger engagement with the first video clip. Similarly, the log (Engagement Max) of the second clip negatively correlated with the reaction time for “angry” (*r* = −0.262, *p* = 0.047) and “sad” (*r* = −0.308, *p* = 0.019) conditions, indicating that those children who were faster at recognizing “anger” and “sad” expressions tended to engage more with the second video clip. Last but not least, children’s reaction times for dynamic facial emotion recognition tasks were positively correlated among different expressions, indicating a consistency in the individual’s speed of responses.

## 4. Discussion

Using an empathy quotient and two computerized tasks, the present study investigated the development of dynamic facial emotion recognition in 5- to 9-year-old children and adults and explored the associations among the individuals’ empathy score, motor empathy, and dynamic emotion expression recognition. We obtained several noteworthy findings. For dynamic facial emotion recognition, recognizing “happy” was the easiest; the 5- to 6-year-olds’ accuracy was just as good as the adults’. On the other hand, “fearful,” “angry,” and “sad” expressions were more challenging, and the accuracy of 5- to 6-year-olds was significantly lower than that of the adults. Moreover, the 5- to 6-year-olds had the longest reaction times across all age expressions. The 7- to 9-year-olds showed the highest spontaneous facial muscle activity for motor empathy, whereas the young adults showed the least. Last but not least, the correlation analysis revealed that the EQ score was positively correlated with the recognition accuracy of “happy” expression for children and was positively correlated with the log (Engagement Max) for adults, meaning that adults who had a higher EQ score also tended to show higher spontaneous facial muscle activity. Below, we will discuss these results and their implications and limitations.

The dynamic emotion recognition results indicated that 5- to 9-year-old children exhibited differential maturation rates [18,19]. This observation supports the notion that the development of emotion recognition is non-uniform. Gao and Maurer (2009) tested 5- to 10-year-olds with various intensities of emotional expressions and found that 5-year-olds were as sensitive as adults in recognizing happiness, even at low intensities. For sadness, 5-year-olds could judge that the face was expressive (i.e., not neutral); however, even at 10, children were still likely to misjudge sadness as fearful. The authors refer to the study by Gao and Maurer (2009) to support the idea that the development of emotion recognition is not uniform. Our finding also echoes the study of Monterosso et al. (2010) [18], which used a dynamic emotion expression (moving from neutral to full intensity) task and reported that sadness and anger were the least accurately recognized expressions among the six basic emotions in 4- to 16-year-old children and adolescents. It is worth mentioning that, in the present study, we found that the accuracy of identifying “sad” expressions might still improve, even in adulthood. In Figure 2, it can be seen that the accuracy in recognizing sadness among parents is slightly higher than that of young adults, with the difference being marginally significant. This finding is consistent with the results of Richoz et al. [11], which indicated that, compared to the recognition of fear, disgust, or surprise, the ability to recognize sadness shows a more gradual improvement until late adulthood, peaking at around 30 years of age. It is possible that as people grow older and gain experience as parents, they become more sensitive to detecting expressions of sadness.

In our study, we utilized the built-in engagement index of the iMotion Affectiva software to approximate motor empathy analysis. This involved detecting patterns of facial muscle activities to estimate the participant’s level of spontaneous motor empathy. Our findings indicated that 7–9-year-old children exhibited the highest level of motor empathy, while college students had the lowest level across all age groups. This suggests that children in this age range may be more attentive to video content and more likely to unconsciously mimic the emotional expressions of the main character in the video. Conversely, college students, who frequently use electronic devices and watch online videos, may be less susceptible to emotional contagion and less likely to engage in spontaneous facial muscle activity due to their exposure to varied content.

We initially anticipated that individual levels of empathy would have a positive correlation with the performance of the dynamic emotional face tasks in both children and adults. However, we only found such a connection in children. We found a significant positive correlation between individual empathy scores and the accuracy of identifying “happy” expressions in children. This suggests that children with higher levels of empathy are better at identifying “happy” expressions, but not other expressions. On the other hand, we did not observe a similar correlation in adults, possibly because adults generally have very high accuracy in recognizing various expressions, particularly happy expressions.

Furthermore, Yan et al. [23] studied the correlation between children’s gaze behavior and empathy. They discovered that children in the high-empathy group were faster at detecting “painful” expressions and were better at evaluating them than children in the low-empathy group. Children with high empathy also tended to give higher ratings to painful expressions compared to children with low empathy. Although we did not use “painful” expressions in our study, we did observe that children who were faster at correctly identifying the “sad”, “happy”, and “angry” expressions (meaning that they performed better) tended to show stronger spontaneous facial muscle activities (log Max engagement) when watching the first video (sad scenes), where the main character was looking sad and crying. Likewise, children who were faster at correctly identifying the “sad” and “angry” expressions tended to show stronger engagement when watching the second video, where the main character was smelling a disgusting smell. These findings are generally consistent with the results of Yan et al. [23].

Notably, the correlation analysis in adults revealed that individuals’ EQ scores positively correlated with certain motor empathy indices such as the log (Engagement Max) of the second (*r* = 0.312, *p* = 0.015) and the third (*r* = 0.405, *p* = 0.001) video clip, indicating that adults with higher EQ scores tended to show stronger spontaneous facial muscle activities when watching the second video (i.e., the main character was smelling a disgusting smell) and the third video (i.e., the main character was watching a funny video). This observation is partially consistent with the ERP study of Choi and Watanuki [22], in which they examined the associations between an individual’s level of empathy and their attention to faces and found that adults with high empathy were more attentive to faces and better at recognizing expressions than adults with lower empathy levels. By detecting the pattern of activities of facial expression muscles as an approximation of spontaneous motor empathy, the built-in engagement index of iMotion Affectiva can be regarded as, in a way, a quantitative index of attention to faces. In our study, adults with higher empathy scores showed stronger attention to faces and were more likely to express their own emotions when watching the short video clips (at least two of them), supporting a meta-analysis showing that stronger facial mimicry responses are positively correlated with level of empathy, but not with facial emotion recognition performance in adults [33].

The current study, in line with the research of Holland et al. [33], did not find a significant link between adults’ EQ score and their ability to recognize dynamic emotional expressions. One explanation could be that adults may have already been very good at recognizing the expressions, leading to a ceiling effect that weakened the correlation with empathy scores. Another possibility is that spontaneous empathy triggered by similar negative personal experiences may have interfered with the recognition of emotional expressions. In a recent study, Israelashvili et al. [43] investigated the relationship between emotion recognition and two empathetic processes: spontaneously felt similarity (having had a similar experience) and deliberate perspective-taking. They found that individuals who had undergone a negative experience identical to the emotional event portrayed in the video stimuli tended to experience greater personal distress after watching the video, which partly explained their reduced accuracy. These findings show that spontaneous empathy driven by similar negative experiences might hinder rather than enhance understanding of others’ emotions.

## 5. Conclusions

In this study, we explored EQ, the recognition of dynamic facial emotional expressions, and motor empathy in 5- to 9-year-old children and adults. We found that children have lower accuracy and longer reaction times in terms of recognizing emotional expressions compared to adults. However, their ability to recognize “happy” expressions is just as good as that of adults, supporting the non-uniform development of emotion recognition. Children with higher EQ scores were more accurate in recognizing “happy” expressions. Adults with higher EQ scores showed higher spontaneous facial muscle activity when watching emotional videos. We acknowledge some limitations in our study, such as our use of the built-in engagement reading of the iMotion Affectiva instead of electromyography (EMG) to detect muscle movements. Future research could explore different motor empathy indicators to further examine the relationship between empathy and emotion recognition.

## Figures and Tables

**Figure 1 sensors-24-04674-f001:**
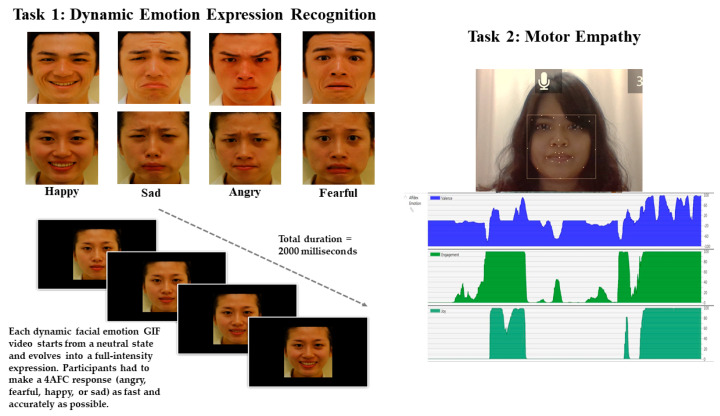
Illustration of the two computerized tasks. Task 1: dynamic facial emotion recognition. The four basic emotion expressions (happy, sad, angry, fear) of male and female face stimuli (**top**) and a sample trial of the dynamic emotion recognition video presented from neutral to full intensity (**bottom**). Task 2: motor empathy. The interface of the i-Motion Affectiva software shows the real-time analysis of a participant’s spontaneous facial expression when watching video clips.

**Figure 2 sensors-24-04674-f002:**
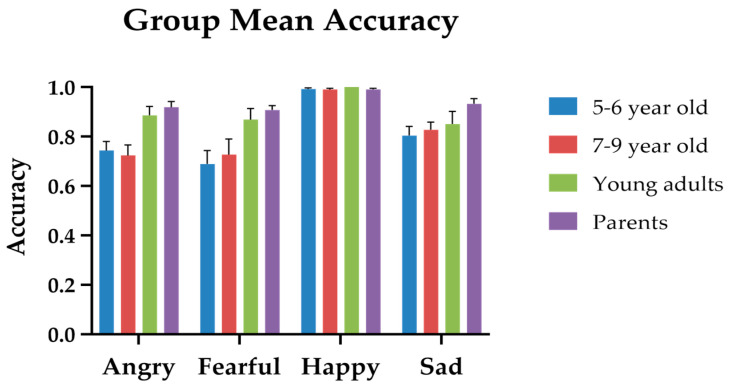
The group mean accuracies for the four emotion expressions in children and adults. The color bars represent different age groups: blue for 5–6-year-olds, red for 7–9-year-olds, green for young adults, and purple for parents.

**Figure 3 sensors-24-04674-f003:**
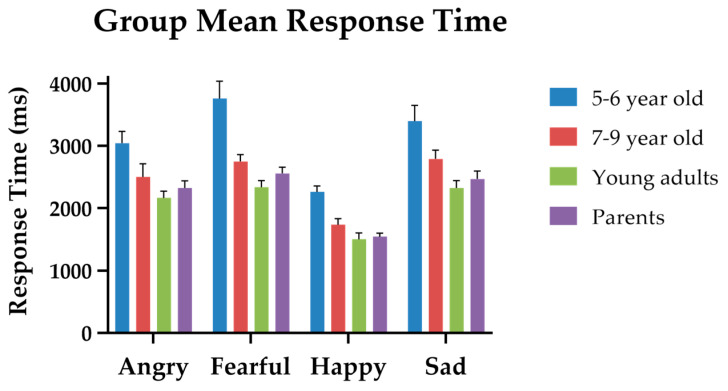
The group mean response times for the four emotion expressions in children and adults. The color bars represent different age groups: blue for 5–6-year-olds, red for 7–9-year-olds, green for young adults, and purple for parents.

**Figure 4 sensors-24-04674-f004:**
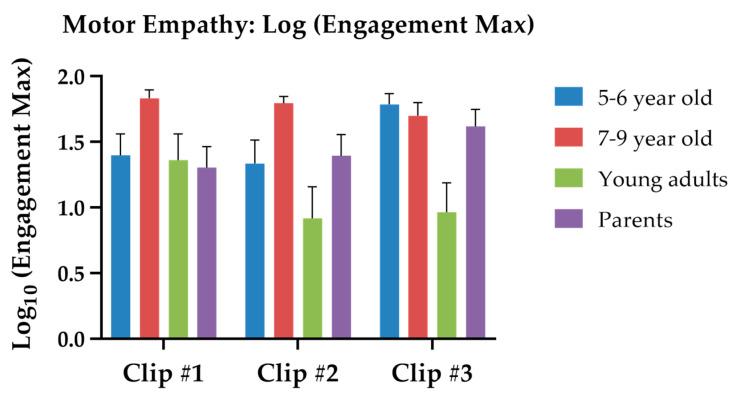
The group mean log maximum engagement of the motor empathy task. The color bars represent different age groups: blue for 5–6-year-olds, red for 7–9-year-olds, green for young adults, and purple for mothers.

**Table 1 sensors-24-04674-t001:** The group mean scores of the Chinese empathy quotient (with standard deviations in parentheses).

Age Groups	Males	Females	All
5–6 years old	18.27 (8.56)	18.56 (6.45)	18.42 (7.36)
7–9 years old	17.54 (6.77)	19.92 (8.88)	18.68 (7.78)
Young adults *	42.17 (17.61)	41.00 10.40)	41.58 (14.15)
Parents *	36.00 (7.95)	48.57 (9.63)	47.89 (9.93)

* The adults and children used different versions of the Chinese empathy quotient.

**Table 2 sensors-24-04674-t002:** Pairwise correlations between EQ score and all performance indices of the dynamic emotion recognition task and the motor empathy task in adults and children (*p*-values are shown in parentheses).

			Dynamic Emotion Recognition Task								Motor Empathy Task		
		EQ Score ^1^	AN_ACC ^2^	FE_ACC ^3^	HA_ACC ^4^	SA_ACC ^5^	AN_RT ^6^	FE_RT ^7^	HA_RT ^8^	SA_RT ^9^	Clip 1 ^10^	Clip 2 ^11^	Clip 3 ^12^
**Adult**	**AN_ACC**	0.167 (0.202)	1	0.091 (0.490)	0.271 (0.037) *	0.111 (0.398)	0.053 (0.685)	0.214 (0.100)	0.191 (0.145)	0.160 (0.226)	0.042 (0.750)	0.039 (769)	0.173 (0.186)
	**FE_ACC**	−0.127 (0.334)	-	1	−0.045 (0.734)	0.429 (0.001) **	−0.031 (0.812)	−0.005 (0.971)	−0.103 (0.431)	0.016 (0.904)	0.085 (0.520)	0.045 (0.731)	0.111 (0.397)
	**HA_ACC**	0.137 (0.295)	-	-	1	−0.020 (0.878)	0.056 (0.670)	0.150 (0.254)	0.147 (0.261)	0.098 (0.461)	−0.083 (0.527)	−0.137 (0.296)	−0.029 (0.823)
	**SA_ACC**	0.194 (0.138)	-	-	-	1	−0.051 (0.699)	−0.050 (0.706)	−0.265 (0.041) *	−0.358 (0.005) **	−0.154 (0.240)	0.111 (0.399)	0.287 (0.026) *
	**AN_RT**	−0.055 (0.679)	-	-	-	-	1	0.530 (0.000) **	0.597 (0.000) **	0.476 (0.000) **	−0.069 (0.603)	−0.040 (0.761)	−0.190 (0.145)
	**FE_RT**	0.146 (0.266)	-	-	-	-	-	1	0.541 (0.000) **	0.747 (0.000) **	−0.020 (0.877)	0.001 (0.996)	−0.125 (0.342)
	**HA_RT**	0.058 (0.659)	-	-	-	-	-	-	1	0.623 (0.000) **	−0.036 (0.786)	−0.060 (0.646)	−0.246 (0.058)
	**SA_RT**	0.223 (0.090)	-	-	-	-	-	-	-	1	0.096 (0.469)	0.133(0.316)	−0.071(0.594)
	**Clip 1**	−0.002 (0.990)	-	-	-	-	-	-	-	-	1	0.448(0.000) **	0.481(0.000) **
	**Clip 2**	0.312 (0.015) *	-	-	-	-	-	-	-	-	-	1	0.577 (0.000) **
	**Clip 3**	0.405 (0.001) **	-	-	-	-	-	-	-	-	-	-	1
**Child**	**AN_ACC**	−0.101 (0.451)	1	0.140 (0.293)	0.087 (0.517)	0.040 (0.765)	−0.015 (0.909)	0.113 (0.412)	−0.170 (0.202)	0.100 (0.455)	−0.053 (0.693)	−0.175 (0.188)	−0.197 (0.139)
	**FE_ACC**	0.185 (0.165)		1	0.104 (0.439)	−0.024 (0.858)	−0.075 (0.573)	−0.179 (0.192)	−0.141 (0.292)	0.202 (0.128)	−0.122 (0.360)	−0.118 (0.377)	−0.200 (0.131)
	**HA_ACC**	0.307 (0.019) *			1	−0.035 (0.795)	0.041 (0.758)	0.050 (0.718)	−0.024 (0.857)	−0.098 (0.465)	−0.014 (0.915)	0.037 (0.783)	−0.147 (0.272)
	**SA_ACC**	−0.076 (0.572)				1	0.131 (0.326)	0.111 (0.422)	0.184 (0.167)	0.190 (0.153)	−0.026 (0.846)	0.079 (0.553)	0.094 (0.483)
	**AN_RT**	−0.041 (0.761)					1	0.567 (0.000) **	0.662 (0.000) **	0.688 (0.000) **	−0.268 (0.042) *	−0.262 (0.047) *	0.142 (0.288)
	**FE_RT**	−0.093 (0.499)						1	0.661 (0.000) **	0.548 (0.000) **	−0.128 (0.353)	−0.086 (0.534)	0.091 (0.510)
	**HA_RT**	−0.094 (0.481)							1	0.517 (0.000) **	−0.276 (0.036) *	−0.140 (0.294)	0.194 (0.145)
	**SA_RT**	0.120 (0.370)								1	−0.268 (0.042) *	−0.308 (0.019) *	0.009 (0.949)
	**Clip 1**	0.081 (0.544)									1	0.680 (0.000) **	0.219 (0.099)
	**Clip 2**	0.153 (0.253)										1	0.170 (0.202)
	**Clip 3**	0.052 (0.697)											1

(* *p* < 0.05, ** *p* = < 0.01). ^1^ EQ scores of Chinese empathy quotient (adult and child versions). ^2–9^ Performance indices of dynamic facial emotion recognition: ^2^ AN_ACC: accuracy for ANGER. ^3^ FE_ACC: accuracy for FEAR. ^4^ HA_ACC: accuracy for HAPPY. ^5^ SA_ACC: accuracy for SAD. ^6^ AN_RT: response time for ANGER. ^7^ FE_RT: response time for FEAR. ^8^ HA_RT: response time for HAPPY. ^9^ SA_RT: response time for SAD. ^10–12^ Performance indices for motor empathy. ^10^ Video Clip 1 log (Engagement Max). ^11^ Video Clip 2 log (Engagement Max). ^12^ Video Clip 3 log (Engagement Max).

## Data Availability

The original datasets (as an Excel file) analyzed during the study have been made publicly available and stored at Open Science Framework via https://osf.io/3g5tw/ (accessed on 14 November 2020).

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
