# Peer review of "Recognition of Dynamic Emotional Expressions in Children and Adults and Its Associations with Empathy"

_sensors, 2024, doi:10.3390/s24144674_

Round 1
Reviewer 1 Report
Comments and Suggestions for Authors
Line 116 indicates 12 female young adults with zero males, however, line 229 presents a statistic for male young adults.
It’s not clear how the clips in 3.3 are selected, which is important since line 349 indicates some sort of correlation.
One significant issue that has to be addressed in this paper is the use of an automated ER system. Let us propose an extreme example where humans are unable to detect a particular emotion X among 4 and they equally confuse it with the 3. Depending on the training approach, it is possible the automated system would produce similar results. In which case we would see a 33% accuracy for detection of this emotion in adults when the premise would suggest the correct answer should be 0%.
Obviously, such an extreme example is unrealistic and unreasonable, but it suggests that the accuracy for automated detection for each emotion should be accounted for when drawing the statistics.
Alternatively, one can suggest that the implicit hypothesis being tested is the ability of an automated system to recognize emotions is similar to humans. However, the paper doesn’t leave an impression that this is the hypothesis.
Overall, the presented numbers (results) do, in fact, resemble the results of previous studies, however, it should be more thoroughly investigated or/and demonstrated that these numbers relate to the same exact problem and drawn from the same, in nature, distribution.
Author Response
[Comment 1] Line 116 indicates 12 female young adults with zero males, however, line 229 presents a statistic for male young adults.
[Answer] Thank you for the comment. In Line 116, we figured that the way we presented in Line 116 might have left an impression that there were no male participants at all. In fact, we have 12 females and 12 males in the young adult group. We have revised the text to clarify this point (see Line 116).
[Comment 2] It’s not clear how the clips in 3.3 are selected, which is important since line 349 indicates some sort of correlation.
[Answer] Thank you for the question. The three video clips were the three stimuli used in the Motor Empathy task; they were selected from YouTube. The most important criterion was that the main character in each clip expressed an easily recognized expression that lasted for about 10 seconds. We originally picked eight clips for pilot testing to make sure that participants could easily understand the emotions in the video. We selected three clips as the final stimuli
- Clip#1 has a main expression of sadness (about 50 seconds), https://www.youtube.com/watch?v=2qaSSdtyvGc&t=463s
- Clip#2 has a main expression of nausea (about 24 seconds),
- https://www.youtube.com/watch?v=oQYPgD3cIP0&t=586s
- Clip#3 has a main expression of happiness (about 31 seconds) https://www.youtube.com/watch?v=4shUUjcjUx8&t=157s
We have added more descriptions in section 2.4.1.
[Comment 3] One significant issue that has to be addressed in this paper is the use of an automated ER system. Let us propose an extreme example where humans are unable to detect a particular emotion X among 4 and they equally confuse it with the 3. Depending on the training approach, it is possible the automated system would produce similar results. In which case we would see a 33% accuracy for detection of this emotion in adults when the premise would suggest the correct answer should be 0%.
Obviously, such an extreme example is unrealistic and unreasonable, but it suggests that the accuracy for automated detection for each emotion should be accounted for when drawing the statistics.
Alternatively, one can suggest that the implicit hypothesis being tested is the ability of an automated system to recognize emotions is similar to humans. However, the paper doesn’t leave an impression that this is the hypothesis.
Overall, the presented numbers (results) do, in fact, resemble the results of previous studies. However, it should be more thoroughly investigated or/and demonstrated that these numbers relate to the same exact problem and are drawn from the same, in nature, distribution.
[Answer] Thank you for the comment. We apologize that we did not fully understand the question. Just for clarification, we used i-Motion Affectiva (a commercial product of an automatic emotion detection program) to assess Motor empathy. We used the log (Engagement Max) as a measure of motor empathy, not the accuracy of emotion expression recognition. We hope this has clarified some of the concerns. Thank you.
Reviewer 2 Report
Comments and Suggestions for Authors
This is a very well-written paper, which describes relevant research and interesting results, comparing the emotional response to facial emotional expressions of 5 to 9 year old children with the response of young adults and parents.
The only comments I have is that a graphical overview of the research framework would help readers understand the project faster.
Also, I find the result that the young adults are the most emotionally expressive and empathic (highest score in fig 3) and motor empathy (fig 4) worth exploring more in-depth in the discussion.
Author Response
[Comment] This is a very well-written paper, which describes relevant research and interesting results, comparing the emotional response to facial emotional expressions of 5 to 9 year old children with the response of young adults and parents.
[Reply] Thank you for your nice words about our study. We appreciate it.
[Comment1] The only comments I have is that a graphical overview of the research framework would help readers understand the project faster.
[Reply] Thank you for the great suggestion. We have added a new graphical abstract.
[Comment 2] Also, I find the result that the young adults are the most emotionally expressive and empathic (highest score in fig 3) and motor empathy (fig 4) worth exploring more in-depth in the discussion.
[Reply] Thank you. Indeed, we have elaborated this point in the Discussion.